# *DAOA* and *APOEε4* as Modifiers of Age of Onset in Autosomal-Dominant Early-Onset Alzheimer’s Disease Caused by the *PSEN1* A431E Variant

**DOI:** 10.3390/ijms26167929

**Published:** 2025-08-16

**Authors:** César A. Valdez-Gaxiola, Frida Rosales-Leycegui, Abigail Gaxiola-Rubio, Sofía Dumois-Petersen, Martha Patricia Gallegos-Arreola, John M. Ringman, Luis E. Figuera

**Affiliations:** 1División de Genética, Centro de Investigación Biomédica de Occidente, Instituto Mexicano del Seguro Social, Guadalajara 44340, Jalisco, Mexico; cesar.valdez2320@alumnos.udg.mx (C.A.V.-G.); frida.rosales9343@alumnos.udg.mx (F.R.-L.); marthapatriciagallegos08@gmail.com (M.P.G.-A.); 2Doctorado en Genética Humana, Centro Universitario de Ciencias de la Salud, Universidad de Guadalajara, Guadalajara 44340, Jalisco, Mexico; 3Maestría en Ciencias del Comportamiento, Instituto de Neurociencias, Centro Universitario de Ciencias Biológicas y Agropecuarias, Universidad de Guadalajara, Guadalajara 45200, Jalisco, Mexico; 4Facultad de Medicina, Universidad Autónoma de Guadalajara, Zapopan 45129, Jalisco, Mexico; abigail.gaxiola2327@alumnos.udg.mx; 5Laboratorio de Ciencias Clínicas, CuValles, Universidad de Guadalajara, Ameca 46600, Jalisco, Mexico; sofia.dumois@academicos.udg.mx; 6Alzheimer Disease Research Center, Department of Neurology, Keck School of Medicine at USC, 1520 San Pablo Street Suite 3000, Los Angeles, CA 90033, USA; john.ringman@med.usc.edu

**Keywords:** *APOE*, age of onset, autosomal-dominant Alzheimer’s disease, early-onset Alzheimer’s disease

## Abstract

While most of the Alzheimer’s disease (AD) cases are sporadic and manifest after age 65 (late-onset AD, LOAD), a subset of patients develop symptoms earlier in life (early-onset, EOAD) due to mutations in the *PSEN1, PSEN2*, or *APP* genes with an autosomal-dominant inheritance pattern (AD-EOAD). In this study, we examined the association between age of onset (AoO) and first clinical manifestation (FCM) with the *APOE* and *DAOA* genotypes, previously described as modifiers of clinical phenotypes in LOAD and EOAD in 88 individuals clinically diagnosed with AD-EOAD due to the *PSEN1* A431E variant (39 females, 49 males). We classified the population according to their genotype (*APOEε2, APOEε3*, and *APOEε4* and *DAOA* G/G, G/A, and A/A) and FCM (cognitive, behavioral, motor, and memory impaired). Memory impairment was the most frequent symptom (51%), followed by motor disturbances (31.8%), cognitive symptoms other than memory (10.4%), and behavioral changes (6.8%). We found a significant association between *APOE* genotype and AoO (*p* < 0.001), with the *APOEε4* allele being linked to a delayed onset (β = 4.04, SE = 1.11, *p* = 0.0003). Similarly, individuals with the *DAOA* rs2391191 A/A genotype showed a significantly later AoO compared to G/G carriers (β = 2.13, SE = 0.96, *p* = 0.0301). No significant association was found between *APOE* or *DAOA* genotypes and FCM. The findings suggest that both the *APOEε4* allele and *DAOA* rs2391191 A/A genotype may act as genetic modifiers of AoO, delaying symptom onset in individuals with AD-EOAD. Further research is needed to elucidate the molecular pathways through which *APOE* and *DAOA* influence AD-EOAD progression.

## 1. Introduction

Alzheimer’s disease (AD) is the most prevalent form of dementia, accounting for approximately 60% to 80% of all diagnosed cases in the elderly population. In the United States alone, an estimated 6.9 million individuals are currently affected. Worldwide, more than 45 million people are living with dementia, and this number is projected to reach 135 million by 2050, primarily influenced by demographic changes and increased life expectancy [1,2]. AD is a progressive neurodegenerative disorder that leads to an irreversible deterioration in cognitive functions, memory, language, behavior, and judgment. Manifesting as cognitive decline, behavioral changes, and a gradual loss of independence, AD places a significant burden on patients, their families, and healthcare systems worldwide [3,4,5]. At the neuropathological level, AD is characterized by the extracellular deposition of amyloid-beta (Aβ) plaques and the intraneuronal accumulation of neurofibrillary tangles (NFTs) composed of hyperphosphorylated tau protein. These hallmark lesions are associated with widespread synaptic dysfunction, neuronal loss, and cortical atrophy. In addition to these classical features, increasing evidence points to the role of chronic neuroinflammation, cerebrovascular alterations, oxidative stress, and mitochondrial dysregulation as critical contributors to the multifaceted pathogenesis of the disease [6,7].

Although most cases of AD manifest after the age of 65—classified as late-onset AD (LOAD)—smaller subsets of individuals develop symptoms at an earlier age, known as early-onset AD (EOAD) [8]. EOAD typically manifests before age 65 and is frequently associated with a more aggressive clinical course, characterized by accelerated cognitive decline and atypical presentations. These often include focal cortical syndromes such as primary progressive aphasia and posterior cortical atrophy, which may complicate diagnosis and lead to initial misclassification. Pathologically, EOAD tends to exhibit severe tau deposition, as evidenced by higher tau-PET signals compared to LOAD [9,10,11,12]. Even though EOAD is frequently associated with Mendelian inheritance, only a small proportion of cases follow an autosomal-dominant pattern [13]. Autosomal-dominant EOAD (AD-EOAD, commonly known as Familial EOAD or Mendelian EOAD) accounts for approximately 10% of all EOAD cases; most of the cases result from variants in the *PSEN1*, *PSEN2*, or *APP* genes [14]. In contrast, LOAD is strongly influenced by the *APOE* gene, particularly the *APOEε4* allele, which increases disease risk and lowers the age of onset (AoO) in a dose-dependent manner [15,16].

The *APOE* gene codes for apolipoprotein E (ApoE), a key regulator of lipid metabolism and neurological function, with significant implications for AD pathogenesis. ApoE is mainly synthesized by astrocytes in the brain. There are three major isoforms—ApoE2, ApoE3, and ApoE4—distinguished by amino acid substitutions at positions 112 and 158: ApoE2 (Cys112, Cys158), ApoE3 (Cys112, Arg158), and ApoE4 (Arg112, Arg158). These isoforms are encoded by the alleles *APOEε2*, *APOEε3*, and *APOEε4*, respectively, whose distribution and biological effects differ substantially. *APOEε3* is the most common allele worldwide (~77%) and is considered neutral regarding AD risk. *APOEε4*, found in ~15% of the general population, produces an isoform with reduced lipid-binding stability and altered receptor interactions. In contrast, *APOEε2*, the rarest allele (~8%), encodes an isoform with enhanced lipid-binding capacity and greater efficiency in Aβ clearance, conferring a protective effect and often delaying AoO. ApoE facilitates lipid transport by interacting with lipoprotein receptors such as LDLr and LRP1, affecting neuronal function, synaptic plasticity, and membrane homeostasis. In the AD context, ApoE plays a crucial role in neural repair, neuroinflammation, and modulation of Aβ clearance, with its isoforms differentially influencing Aβ aggregation and clearance efficiency [17,18].

The *APOEε4* allele is the most well-established genetic risk factor for LOAD, significantly increasing disease susceptibility in a dose-dependent manner. The presence of one *ε4* allele increases the risk of developing AD by 2–4 times, while homozygosity (*ε4*/*ε4*) raises the risk by 8–10 times [15]. In contrast, the *APOEε2* allele exerts a protective effect, reducing disease risk and delaying onset [19] relative to the most common *APOEε3* allele. ApoE4 promotes amyloidogenic processing of the amyloid precursor protein (APP), leading to increased Aβ accumulation, impaired Aβ clearance, and exacerbated tau pathology, neuroinflammation, and blood–brain barrier dysfunction [20,21,22]. Understanding the multifaceted role of ApoE in AD pathogenesis is critical for the development of targeted therapeutic strategies to mitigate disease progression [22].

*DAOA* (also known as *G72*) encodes a protein that modulates the activity of D-amino acid oxidase (DAAO), an enzyme responsible for the degradation of D-serine. This amino acid serves as a potent co-agonist of N-methyl-D-aspartate receptors (NMDARs), which are critical for synaptic plasticity, learning, and memory [23,24].

The enzymes involved in the metabolism of D-amino acids have been associated with the pathophysiology of AD and schizophrenia [25,26]. In this context, the variant rs2391191 involves a guanine-to-adenine substitution at position 6231, resulting in an amino acid change from arginine to lysine at codon 30. This substitution has been reported to affect the structure and function of the DAOA protein [27].

The disruption of glutamate-mediated neurotransmission is considered a key mechanism in the pathogenesis of AD. L-serine is converted into D-serine by the enzyme serine racemase (SR), which is expressed in both astrocytes and presynaptic neurons. D-serine can be released into extracellular space by the alanine–serine–cysteine transporter (ASC-1) and ASCT-1 transporters. In contrast, L-serine is released by astrocytes and taken up by neurons via the same transporters, but in the reverse direction. D-serine is subsequently degraded by DAAO, an enzymatic activity that can be inhibited by DAOA, particularly when DAOA is localized to the outer mitochondrial membrane [28,29].

It has been proposed that the rs2391191 variant may indirectly reduce D-serine levels, thereby modulating the activation of NMDAR, which is known to be aberrantly upregulated in AD. This regulatory mechanism could potentially contribute to cognitive preservation and delay the onset of symptoms in EOAD [30].

Presenilin 1 (PS1, encoded by the gene *PSEN1*) is a crucial component of the γ-secretase complex, a multi-subunit protease involved in the intramembrane cleavage of several type I transmembrane proteins, most notably APP. Through this role, PS1 modulates the production of Aβ peptides [31,32]. Mutations in the *PSEN1* gene are the most frequent cause of AD-EOAD, often leading to altered γ-secretase function, which increases the relative production of longer Aβ species [33,34]. The A431E variant is located within the ninth transmembrane domain of PS1 and has been identified in Mexican families. This variant is associated with an especially aggressive clinical course, including very early symptom onset, rapid cognitive decline, and frequent neuropsychiatric and motor disturbances [35]. Functional analyses suggest that the A431E variant significantly impacts the structural conformation and enzymatic activity of γ-secretase, resulting in elevated Aβ42 levels [36].

Since 1999, evidence has suggested that a significant proportion of individuals with AD-EOAD in the state of Jalisco, Mexico, carry the *PSEN1* A431E pathogenic variant due to a founder effect. This phenomenon has been particularly observed in the mestizo population of Altos de Jalisco, a region characterized by mixed Spanish and indigenous ancestry [35,37].

The A431E variant in *PSEN1* was first described by Rogaeva et al. [38] in five patients of unspecified origin. Subsequent investigations identified this variant in nine unrelated families native to Jalisco [37], strengthening the hypothesis of a common ancestral origin. In 2006, Murrell et al. reported 15 additional unrelated cases carrying the same variant [35], further reinforcing its association with a regional founder effect. The *PSEN1* A431E variant is currently classified as pathogenic [35]. The clinical characterization of affected individuals revealed an AoO mean of 42.5 ± 3.9 years. Notably, there was marked clinical heterogeneity in the first clinical manifestations (FCM), with frequent reports of spastic paraparesis, language disturbances, and neuropsychiatric symptoms [36].

Previous studies have reported that the *APOE* genotype was associated with an effect on AoO [39,40,41] and clinical manifestations [42,43,44,45] in EOAD patients.

In the Colombian kindred with the *PSEN1* E280A variant, studies have shown that the *APOEε4* and *APOEε2* alleles are associated with earlier and later AoO, respectively [40,46]. A larger study involving persons with various AD-EOAD mutations suggested only a subtle effect of earlier onset in association with the *APOEε4* allele that was not statistically significant [47]. Interestingly, we previously published that patients with AD-EOAD and EOAD (with and without a known genetic cause) had a delay in the AoO due to the *APOEε4* allele [48]. In the present study, we aimed to investigate the association between AoO and FCM with the *APOE* and *DAOA* genotypes in patients harboring the *PSEN1* A431E pathogenic variant.

## 2. Results

### 2.1. Sociodemographic Information

A total of 88 index cases of AD-EOAD caused by the A431E variant in *PSEN1* were evaluated. Of these cases, 49 (55.7%) were male and 39 (44.3%) were female. The average AoO was 41.81 years, ranging from 34 to 49 years (Table 1).

FCM were categorized into four main domains based on their frequency: cognitive (excluding memory), behavioral, motor, and memory-related symptoms. Within the cognitive domain, a total of nine cases (10.4%) were documented, including aphasia, anomia, dysgraphia, dyslalia, verbal perseveration, and disorientation. Behavioral symptoms were observed in six individuals (6.8%) and included irritability, depression, and aggressiveness. Motor symptoms were present in 28 cases (31.8%), predominantly characterized by gait disturbances, as well as dysarthria and apraxia. Finally, memory impairment was the most frequently reported symptom, present in 45 individuals (51%). This classification enabled a more structured analysis of symptom distribution and facilitated comparisons across domains in relation to genetic and clinical variables.

### 2.2. Age of Onset, DAOA, and APOE Genotype

The most frequent *APOE* genotype was *ε3*/*ε3*, found in 64 individuals (72.73%). The *ε3*/*ε4* genotype was identified in 10 cases (11.36%), *ε2*/*ε3* in 7 cases (7.96%), *ε2*/*ε2* in 5 cases (5.69%), *ε2*/*ε4* in 1 case (1.13%), and *ε4*/*ε4* in 1 case (1.13%). In terms of allele frequency, *ε3* was the most prevalent (82.38%), followed by *ε2* (10.22%) and *ε4* (7.4%). For analytical purposes, individuals were grouped based on the presence of the *APOEε2* or ε4 alleles. The *APOEε2+* subgroup included patients carrying at least one ε2 allele, specifically those with *APOEε2*/*ε2* and *APOEε2*/*ε3* genotypes. The *APOEε4+* subgroup included those carrying at least one ε4 allele: *APOEε2*/*ε4*, *APOEε3*/*ε4*, and *APOEε4*/*ε4* genotypes. This grouping approach allowed us to explore potential allele-specific effects on AoO while acknowledging the limitations imposed by the relatively small number of *APOEε2* and *APOEε4* homozygotes in the cohort. Regarding the *DAOA* rs2391191 genotypes, G/G was observed in 35 individuals (39.77%), G/A in 32 individuals (36.36%), and A/A in 21 individuals (23.87%). Allele frequency analysis showed that the G allele was the most prevalent (57.95%), followed by the A allele (42.05%).

We examined the distribution of *APOE* genotypes and their association with clinical characteristics in our *PSEN1* variant carrier cohort (*n* = 88). The mean AoO significantly differed among *APOE* genotype groups (*H* = 18.0853, *p* = 0.00012), with *APOEε2+* individuals presenting the earliest mean AoO (39.2 ± 3.1 years), followed by *APOEε3* (41.6 ± 3.7 years), and *APOEε4+* carriers showing the latest mean onset (45.8 ± 2.5 years).

A Mann–Whitney U test showed no significant difference in AoO between sexes (W = 903, *p* = 0.661). Likewise, the predominant FCM did not vary significantly across *APOE* genotypes (*p* = 0.7248), with memory impairment being the most frequent presentation in all groups. The results reveal a significant difference in the AoO across different *APOE* genotypes. A *p*-value less than 0.001 indicates a strong difference between AoO and *APOE* genotypes; however, no significant difference was observed between the FCM and *APOE* genotypes (Table 1).

We further assessed the potential influence of *DAOA* genotypes on the clinical phenotypes within the *PSEN1* A431E variant carrier cohort. A statistically significant difference in AoO was detected among genotype groups (*H* = 10.334, *p* = 0.0057). Individuals with the A/A genotype exhibited a later mean AoO (44.1 ± 3.2 years) compared to G/G (41.4 ± 3.8 years) and G/A carriers (40.7 ± 3.9 years).

FCM did not differ significantly across genotype groups (*p* = 0.5284), with memory symptoms remaining the most prevalent initial presentation. The results reveal a significant difference in the AoO across different *DAOA* genotypes. A *p*-value of <0.05 indicates a strong difference between the AoO and *DAOA* genotypes; however, no significant association was observed between the FCM and *DAOA* genotypes (Table 2).

### 2.3. Effect of APOE and DAOA Genotype on Age of Onset

Due to the significant findings from the non-parametric analysis, a simple linear regression was performed to quantify the impact of *APOE* and *DAOA* genotypes on AoO (Table 3 and Table 4). The results confirmed a significant association between *APOE* and *DAOA* genotypes and AoO, highlighting a potential role in modulating disease onset (Figure 1 and Figure 2). In contrast, no regression analysis was performed for the FCM, because Fisher’s exact test did not indicate a significant difference.

To evaluate the impact of genetic variants on AoO in AD-EOAD, we performed a multiple linear regression including *APOEε2*, *APOEε4*, and genotypes of the *DAOA* rs2391191 variant as predictors. The model was statistically significant (F(4, 86) = 7.911, *p* = 1.88 × 10^−5^).

As shown in Table 5, the presence of the *APOEε4* allele was significantly associated with a later AoO (β = 4.04, SE = 1.11, *p* = 0.000332), whereas the *APOEε2* allele was not significantly associated with AoO (*p* = 0.14). Regarding *DAOA*, individuals homozygous for the A allele (A/A genotype) exhibited a significantly delayed onset compared to the reference genotype (G/G) (β = 2.13, SE = 0.96, *p* = 0.0301). The heterozygous G/A genotype did not show a statistically significant association (*p* = 0.39).

As shown in Table 6, the presence of the *APOEε4* allele was significantly associated with a later AoO (β = 6.24, SE = 1.83, *p* = 0.00105), whereas the *APOEε2* allele did not show a significant association (*p* = 0.867). For *DAOA*, individuals with the A/A genotype exhibited a significantly delayed onset compared to the reference genotype G/G (β = 3.00, SE = 1.07, *p* = 0.0063). No significant association was observed for the heterozygous G/A genotype (*p* = 0.99). Interaction terms between *APOE* alleles and *DAOA* genotypes were included in the model but did not reach statistical significance.

These results suggest that both the *APOEε4*+ and the *DAOA* rs2391191 A/A genotype may independently modulate the clinical presentation of AD-EOAD by delaying the AoO (Figure 3).

## 3. Discussion

Understanding the genetic factors that influence the AoO in AD-EOAD is critical for unraveling disease mechanisms. While *APOE* and *PSEN1* variants have well-documented effects on AD risk and progression, the interaction of these genes with other loci, such as *DAOA*, remains less clear.

Our findings reveal a significant association between *APOE* and *DAOA* genotypes and the AoO in individuals with AD-EOAD caused by the *PSEN1* A431E variant. The presence of the *APOEε4* allele was associated with a delayed clinical onset, a result that contrasts with well-established observations in LOAD, where *APOEε4* is typically linked to earlier disease onset and increased severity [16,41,49,50,51,52].

Several studies have reported the heterogeneous effects of *APOEε4* in EOAD and AD-EOAD populations [39,41,44,48,53,54], suggesting that its modulatory role may be context-dependent. This divergence from the LOAD findings supports the hypothesis that the influence of APOE on AD pathophysiology may vary according to age at onset and underlying genetic background.

The role of *APOE* in modulating the clinical trajectory of AD-EOAD has been a matter of ongoing debate. Initial studies, such as the one by Lendon et al. (1997), suggested that *APOE* genotypes had no significant impact on AoO in *PSEN1* E280A carriers [55]. However, more recent work has challenged this view. Vélez et al. (2016), for example, reported a modifier effect of *APOE*ε2, with significantly delayed onset among *PSEN1* E280A carriers [40]. Similarly, Langella et al. (2023) showed that cognitive decline in AD-EOAD is accelerated in *APOE*ε4 carriers and attenuated in *APOE*ε2 carriers, underscoring the allele-specific influence of *APOE* on disease expression, even in the presence of fully penetrant monogenic mutations [56].

Our observation of a delayed onset in *APOEε4* carriers with the *PSEN1* A431E variant is particularly intriguing. It echoes the results by De Luca et al. (2016), who reported a paradoxical effect of *APOE*ε4: accelerating disease onset in LOAD yet delaying it in certain EOAD contexts [39]. These unexpected results suggest that the modulatory role of *APOE* is not uniform across AD subtypes and may interact with specific genetic backgrounds—such as *PSEN1* variants—to alter its phenotypic expression. Additionally, Smits et al. (2015) highlighted that *APOE*ε4-negative individuals exhibited more rapid decline in non-memory cognitive domains, suggesting that *APOE* may influence not only AoO but also domain-specific trajectories of cognitive impairment [44]. Our results align with this growing recognition that *APOE*’s impact in AD extends beyond a unidimensional effect on AoO.

In the context of AD-EOAD caused by the *PSEN1* A431E, the protective effect of *APOEε2* appears to be non-significant, in contrast with observations in persons carrying the E280A *PSEN1* variant which showed a delayed AoO associated with the *APOEε2* allele [40]. The delayed onset observed in *APOEε4* carriers challenges the conventional understanding of *APOEε4* as a risk factor for accelerated cognitive decline. One possible explanation is that *APOE* may exert its effects later in the disease process, after significant neurodegeneration has already occurred. The delayed AoO in *APOEε4* carriers may also reflect compensatory mechanisms, such as neuroinflammatory responses and altered lipid metabolism [56,57], which initially mitigate the pathological impact of the *PSEN1* A431E variant.

The paradoxical findings of the present study can be explained by the concept of antagonistic pleiotropy. According to this model, the effects of a gene can be beneficial in some life stages and detrimental in others [58]. Research suggests that *APOEε4* carriers may exhibit cognitive advantages in youth and early adulthood, including higher IQ scores [59], better performance on neuropsychological tests assessing attention and memory [60,61], and increased mental vitality. Furthermore, *APOEε4* has been linked to personality traits such as sociability and positive emotionality, which may provide adaptive benefits during early life [62].

This compensatory mechanism enables *APOEε4* carriers to exhibit relatively high cognitive performance early in life, even in the presence of subclinical neurocognitive changes associated with AD [63]. However, as *APOEε4* carriers age, this compensatory process begins to fail. When the pathological burden of AD reaches a critical threshold, compensatory mechanisms are no longer sufficient to sustain cognitive function, resulting in a later onset of symptoms compared to non-carriers [64]. While *APOEε4* enhances microglial activation, this effect becomes detrimental in LOAD, where prolonged or dysregulated activation impairs Aβ clearance and exacerbates neuroinflammation [65,66].

Our findings support the hypothesis that the *DAOA* rs2391191 variant may act as a genetic modifier, capable of influencing the phenotypic expression of AD-EOAD. Our findings contribute additional evidence to the role of common variants in shaping the heterogeneous clinical trajectories observed in individuals with AD-EOAD [40].

At the molecular level, DAOA may exert a modifying effect by indirectly regulating NMDAR signaling by modulating D-serine levels. Previous studies have reported that downregulation or reduced functional activity of DAOA—which may occur in individuals homozygous for the A allele—could lead to more stable D-serine levels, resulting in a more balanced activation of NMDARs. This mechanism may delay synaptic dysfunction in AD-EOAD (see Figure 4) [27,29,30,67,68].

Moreover, the lack of an additive effect with *APOEε4* suggests that *DAOA* operates through a distinct biological pathway, independent of lipid metabolism and cholesterol transport, mechanisms predominantly associated with ApoE.

Taken together, these findings support a model in which *APOEε4* may be beneficial in early life but becomes detrimental in aging due to its effects on lipid metabolism, cellular stress responses, and Aβ aggregation. Further research is needed to dissect the precise molecular pathways through which *APOE* and *DAOA* influence AD-EOAD and to explore potential therapeutic strategies.

## 4. Materials and Methods

This study included 88 patients who were diagnosed clinically with AD-EOAD and confirmed to carry the *PSEN1* A431E variant (39 females and 49 males). The patients were evaluated at the División de Genética at the Centro Médico Nacional de Occidente—Instituto Mexicano del Seguro Social (IMSS), in Guadalajara, Jalisco, Mexico, between 2012 and 2025. Genealogy and clinical data were also gathered from unaffected family members.

Primary caregivers of the patients were interviewed to determine AoO and to document the FCM. A thorough and structured clinical history was collected for each patient. Next, molecular analysis was performed on the probands to confirm the genetic diagnosis.

Written informed consent was obtained from the primary caregiver or legal representative of each participant prior to their inclusion in the study.

### 4.1. DNA Extraction and Sanger Sequencing

DNA was extracted from peripheral blood samples using the salting-out method [69]. For Sanger sequencing, the amplification was targeted specifically to exon 12 of the *PSEN1* gene, where the variant A431E (c.1292C>A, rs63750083) is located. After amplifying the region of interest via PCR, the amplification product was purified using ExoSAP-IT™ Express reagent (Applied Biosystems™, Foster City, CA, USA) to remove any remaining primers and unincorporated nucleotides. The sequencing reaction was carried out using the BigDye Terminator v3.1 Cycle Sequencing Kit (Applied Biosystems™, Foster City, CA, USA). Finally, the sequencing product was purified with the BigDye™ XTerminator Purification Kit (Applied Biosystems™, Foster City, CA, USA), according to the manufacturer’s specifications, to ensure a clear and contaminant-free sequence.

### 4.2. APOE and DAOA Genotyping by Real-Time PCR

*APOE* alleles were inferred from single-nucleotide polymorphisms (SNPs) rs7412 (C____904973_10) for *APOEε2* allele and rs429358 (C___3084793_20) for *APOEε4* allele, which were genotyped using real-time PCR. *APOEε4* genotyping for these two SNPs was performed using a TaqMan™ assay (Applied Biosystems™, Foster City, CA, USA). Each reaction contained 20–50 ng of DNA, 2X TaqMan™ Genotyping Master Mix, and specific TaqMan™ probes labeled with VIC and FAM fluorophores to detect the rs7412 and rs429358 variants. The cycling conditions included an initial enzyme activation step at 95 °C for 10 min, followed by 40 cycles of denaturation at 95 °C for 15 s and annealing/extension at 60 °C for 1 min.

Genotyping of the *DAOA* rs2391191 (C__16000591_10) variant was also performed using real-time PCR with TaqMan™ technology, following the same reaction setup and thermal cycling conditions as described for the *APOE* SNPs.

The fluorescence signals were analyzed using Applied Biosystems’ proprietary software. To assign a genotype to each patient, both the detection and intensity of the fluorophores were evaluated. For quality control, a subset of samples was genotyped in duplicate, and any ambiguous results were resolved by repeating the assay.

### 4.3. Dementia Diagnosis

The diagnosis of dementia was established through a multidisciplinary consensus involving a board-certified neurologist and two physicians specializing in dementia. The diagnostic process included a comprehensive neurological assessment, an in-depth clinical examination, and a detailed review of relevant medical and functional history before the dementia evaluation. Additionally, cognitive and behavioral changes were assessed through informant interviews to supplement clinical observations.

The classification of dementia followed internationally recognized guidelines, with cognitive impairment initially screened using the Mini-Mental State Examination (MMSE). For AD, the diagnosis was based on the criteria established by the National Institute of Neurological and Communicative Disorders and Stroke (NINCDS) in collaboration with the Alzheimer’s Disease and Related Disorders Association (ADRDA). These criteria ensure diagnostic accuracy by integrating clinical, cognitive, and functional assessments.

### 4.4. Ethics

This study was conducted in strict compliance with the Regulations of the General Health Law on Health Research [70], as well as the ethical principles outlined in the Declaration of Helsinki, last updated in October 2024 by the World Medical Association. Additionally, the study adhered to both national and international guidelines for best practices in clinical research [71].

According to the classification established in the Regulations of the General Health Law on Health Research [70], this study falls under type I research, meaning that it poses no risk to the participants. This study involved the collection of DNA samples through venipuncture, a standard, minimally invasive procedure. All samples were anonymized using a unique folio number and securely stored. The corresponding data were systematically recorded in a protected database, ensuring strict confidentiality and compliance with data protection regulations.

### 4.5. Statistical Analysis

Descriptive and comparative analyses of both sociodemographic and clinical data were conducted using RStudio v2024.12.1+563. Data visualization was performed through graphical representations generated in the same software.

To compare categorical variables, Fisher’s exact test or the chi-square test was applied, depending on data distribution and expected frequencies. To assess the normality of the age of symptom onset variable, a Shapiro–Wilk test was performed. The Kruskal–Wallis test was used to evaluate significant differences between the AoO, the FCM, and *APOE* and *DAOA* genotypes. To assess the extent of the effect of *APOE* and *DAOA* genotypes on the AoO, a simple linear regression analysis was performed. To further assess the extent of the effect of *APOE* and *DAOA* genotypes on AoO, a multiple linear regression analysis was conducted, including both genotypes as predictors.

## 5. Conclusions

This finding contrasts with previous evidence linking the *APOEε4* allele to an increased risk and earlier onset of LOAD, indicating that its effect may differ in populations with a strong genetic background, such as this cohort. This result is particularly intriguing, because *APOEε2* has been widely associated with a protective role against AD in sporadic cases.

The variability in AoO across genotypes underscores the complex role of *APOE* in modulating disease progression, potentially through mechanisms involving lipid metabolism, neuroinflammation, and amyloid processing. These findings suggest that *APOEε4* may not always confer an earlier onset in certain genetic backgrounds, warranting further investigation into its potential modifying effects. However, these findings align with the hypothesis of antagonistic pleiotropy, where *APOEε4* provides early-life advantages at the cost of late-life neurodegeneration. Moreover, the *DAOA A/A* genotype was found to delay the age of symptom onset by approximately 2 years, suggesting a possible protective influence. This effect may be mediated by DAOA’s role in modulating D-Serine levels, a key co-agonist of the NMDAR involved in glutamatergic neurotransmission, synaptic plasticity, and neurotoxicity.

### Limitations and Future Directions

Despite the significance of our findings, several limitations must be acknowledged. First, our study is limited by the relatively small sample size, which may impact the generalizability of our results. Additionally, the cohort consists of individuals from a genetically homogeneous Mexican founder population, all carrying the *PSEN1* A431E variant. While this homogeneity likely enhances our ability to detect genotype–phenotype associations by reducing background genetic noise, it also restricts the external validity of our results. Specifically, the observed associations may not be generalizable to individuals of different ethnic backgrounds, to sporadic forms of AD, or to carriers of other *PSEN1* variants. For example, we observed that there is only one *APOEε4/ε4* carrier in our cohort. This limits our ability to explore or interpret potential dose-dependent effects of the ε4 allele, which have been reported in other contexts to be associated with an earlier age at onset and increased disease severity. Larger cohorts are necessary to validate our findings and further investigate genotype-specific effects. However, as far as we know, this is the second largest population worldwide with AD-EOAD, which is considered an ultra-rare disease. Second, our study primarily focuses on AoO and does not comprehensively assess other clinical parameters, such as cognitive trajectories or biomarker progression. Longitudinal studies that incorporate multimodal biomarker assessments will be essential to elucidate the broader implications of the *APOE* and *DAOA* genotypes in AD-EOAD.

Additionally, while we propose potential mechanisms underlying the differential effects of *APOE* and *DAOA* genotypes in AD-EOAD, our study lacks the direct molecular or neuropathological data necessary to confirm these hypotheses. Future research should integrate functional studies examining these pathways, particularly in the context of *PSEN1* variants, to better understand their combined impact on disease progression.

## Figures and Tables

**Figure 1 ijms-26-07929-f001:**
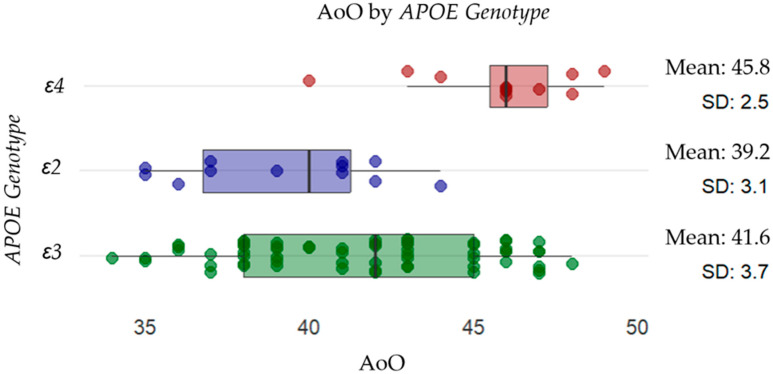
Distribution of AoO across different *APOE* genotypes. Individual data points are displayed as scattered dots, with the mean AoO and standard deviation (SD) indicated for each genotype. The results suggest that individuals with the *APOEε4* allele (red) (mean: 45.8, SD: 2.5) have a later onset than those with the *APOEε2* allele (purple) (mean: 39.2, SD: 3.1) and *APOEε3* (green) (mean: 41.6, SD: 3.7).

**Figure 2 ijms-26-07929-f002:**
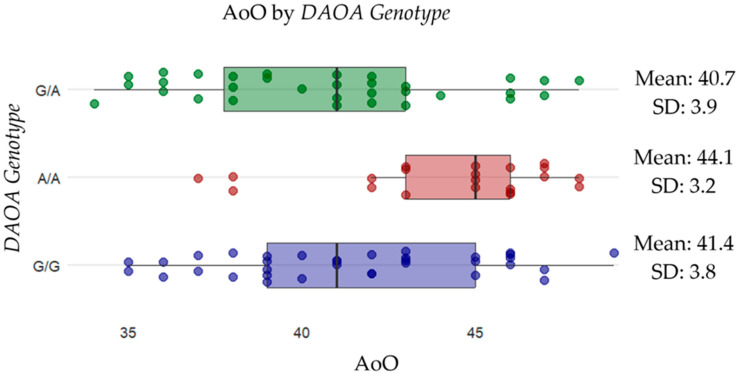
Distribution of AoO across different *DAOA* genotypes. Scattered dots represent individual data points, with the mean AoO and standard deviation (SD) shown for each genotype. The results suggest that individuals with the *A/A* genotype (red) (mean: 44.1, SD: 3.2) have a later onset compared to *G/A* (purple) (mean: 39.2, SD: 3.1) and *APOEε3* (green) (mean: 41.6, SD: 3.7).

**Figure 3 ijms-26-07929-f003:**
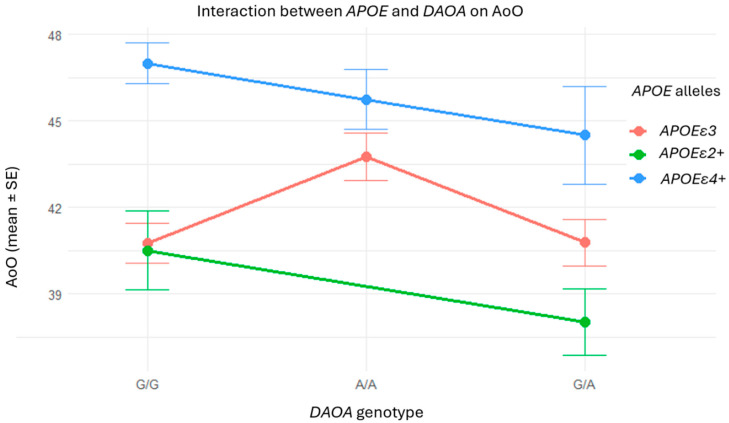
Interaction between *DAOA* and *APOE* genotypes on AoO: no consistent additive or synergistic effects. Although patients with the A/A genotype or *APOEε4+* individually tended to exhibit later onset, the visualization does not reveal any consistent additive or interaction patterns between these variants. The absence of clear trends supports the statistical results, indicating no synergistic effects between *APOE* and *DAOA* in this cohort.

**Figure 4 ijms-26-07929-f004:**
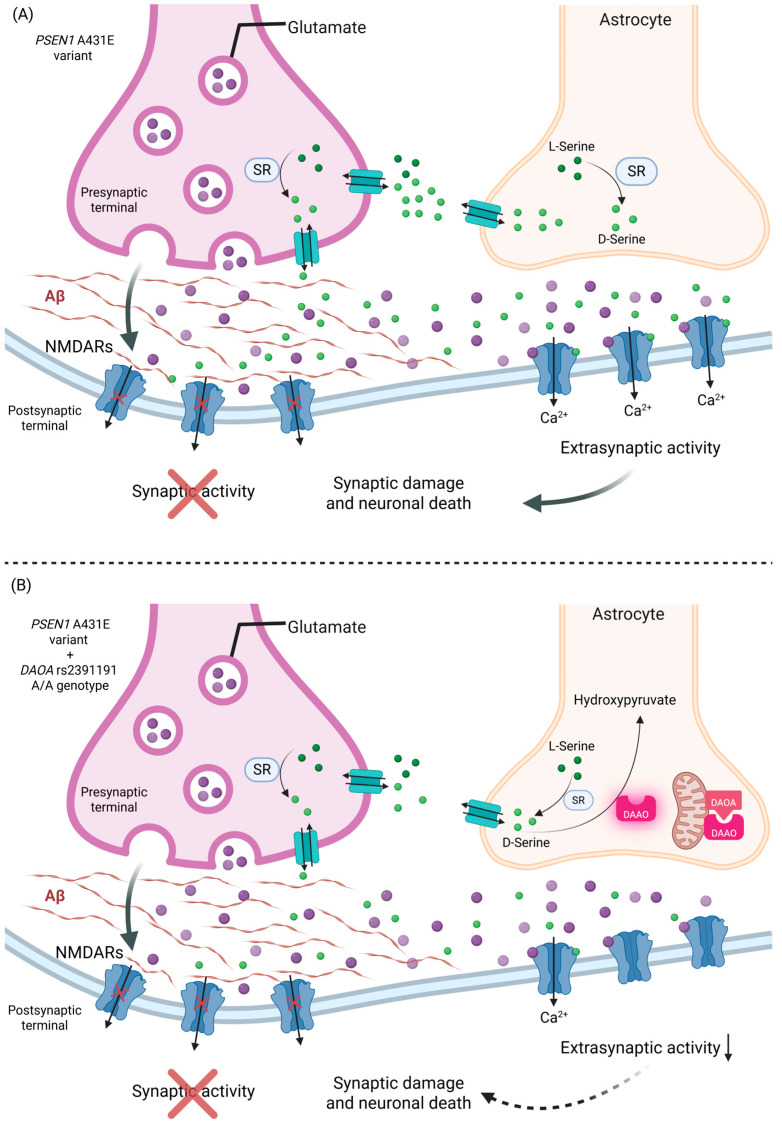
Effect of the *DAOA* rs2391191 A/A genotype on glutamatergic signaling in the presence of the *PSEN1* A431E variant. The *PSEN1* A431E variant promotes the accumulation of Aβ at the synaptic cleft, blocking the normal synaptic activity and shifting the activation of NMDARs to the extrasynaptic space, where excitotoxic pathways associated with synaptic damage and neuronal death are enhanced. (**A**) In the absence of the *DAOA* rs2391191 A/A genotype, D-serine generated by SR is normally degraded due to DAAO inactivation by DAOA, enabling the activation of extrasynaptic NMDARs and favoring neuronal damage. (**B**) In the presence of the *DAOA* rs2391191 A/A genotype, it could decrease the affinity of DAOA for DAAO, allowing DAOA to remain active and degrade D-serine to hydroxypyruvate, reducing the activation of extrasynaptic NMDARs and attenuating excitotoxic damage. This mechanism could contribute to a later onset of symptoms in patients with AD-EOAD in A/A rs2391191 carriers. Created in BioRender. https://BioRender.com/93xxevk (accessed on 30 June 2025).

**Table 1 ijms-26-07929-t001:** Clinical and demographic characteristics by *APOE* genotype in *PSEN1* A431E variant carriers.

APOE	APOEε2+	APOEε3	APOEε4+	Total
Patients	12	64	12	88
AoO (years) ^a^	39.2 ± 3.1	41.6 ± 3.7	45.8 ± 2.5	41.8 ± 3.89
Mean ± SD				
Min–Max	35–44	34–48	40–49	34–49
*H*	18.0853			
*p*-value	0.00012 ***			
Sex ^b^				
Male	5	35	9	49
Female	7	29	3	39
*W*	903			
*p*-value	0.6611			
FCM ^c^				
Memory	7	32	6	45
Motor	2	21	5	28
Behavioral	2	4	0	6
Cognitive	1	7	1	9
*p*-value	0.7248			

^a^ Kruskal–Wallis test, ^b^ Mann–Whitney U test, and ^c^ Fisher’s exact test. Statistical significance was defined as *p <* 0.001 (***).

**Table 2 ijms-26-07929-t002:** Clinical and demographic characteristics by *DAOA* genotype in *PSEN1* A431E variant carriers.

DAOA	G/G	G/A	A/A	Total
Patients	35	32	21	88
AoO (years) ^a^	41.4 ± 3.8	40.7 ± 3.9	44.1 ± 3.2	41.8 ± 3.89
Mean ± SD				
Min–Max	35–49	34–48	37–48	34–49
*H*	10.334			
*p*-value	0.0057 **			
Sex				
Male	15	12	12	49
Female	20	20	9	39
FCM ^c^				
Memory	19	15	11	45
Motor	10	10	8	28
Behavioral	4	1	1	6
Cognitive	2	6	1	9
*p*-value	0.5284			

^a^ Kruskal–Wallis test, ^c^ Fisher’s exact test. Statistical significance was defined as *p <* 0.01 (**).

**Table 3 ijms-26-07929-t003:** Simple linear regression of *APOEε2* and *APOEε4* alleles with AoO.

	Value	Standard Error	T Value	*p*-Value
(Intercept)	41.562	0.444	93.617	2 × 10^−16^ ****
*APOE* *ε2+*	−2.312	1.117	−2.116	0.0415 *
*APOE* *ε4+*	4.188	1.117	3.752	0.00035 ***
Residual standard error: 3.552				
R-square adjusted: 0.1817				
F statistic: 10.66 with 85 grades of freedom				
*p*-value of F statistic: 7.423 × 10^−5^ ********				

*APOEε2* (−2.312): Carriers of the *APOEε2* allele had an earlier mean AoO, approximately 2.312 years younger than the reference group (*APOEε3/ε3*). *APOEε4* (4.188): Individuals carrying one or two *APOEε4* alleles exhibited a later mean AoO, approximately 4.188 years older than *APOEε3/ε3* homozygotes. Statistical significance was defined as *p* < 0.05 (*)*, p <* 0.001 (***) and *p <* 0.0001 (****).

**Table 4 ijms-26-07929-t004:** Simple linear regression of *DAOA* rs2391191 genotypes with AoO.

	Value	Standard Error	T Value	*p*-Value
(Intercept)	41.4286	0.6308	65.681	2 × 10^−16^ ****
*A/A*	2.7143	1.0300	2.635	0.00999 **
*G/A*	−0.7098	0.9127	−0.778	0.43889
Residual standard error: 3.732				
R-square adjusted: 0.09666				
F statistic: 10.8 with 85 grades of freedom				
*p*-value of F statistic: 0.004948 **				

*DAOA* A/A homozygous (2.7): Individuals homozygous for the A allele of *DAOA* exhibited a later mean AoO, approximately for 2.7 years than G/A and G/G patients. Statistical significance was defined as *p <* 0.01 *(***) and *p <* 0.0001 (****).

**Table 5 ijms-26-07929-t005:** Multiple linear regression of *APOE* and *DAOA* with AoO.

	Value	Standard Error	T Value	*p*-Value
(Intercept)	41.2474	0.6258	65.908	2 × 10^−16^ ****
*APOE* *ε2+*	−1.634	1.1012	−1.484	0.141643
*APOE* *ε4+*	4.0359	1.1077	3.752	0.000332 ***
*DAOA G/A*	−0.7268	0.8368	−0.869	0.387615
*DAOA A/A*	2.1267	0.964	2.206	0.030134 *
Residual standard error: 3.42				
R-square adjusted: 0.2411				
F statistic: 7.911 with 2 and 86 grades of freedom				
*p*-value of F statistic: 1.879 × 10^−5^ ****				

A multiple linear regression was performed to examine the effects of *APOEε4*, and *DAOA* rs2391191 genotypes on AoO in AD-EOAD. The reference category for *DAOA* is the G/G genotype. Shown are the estimated regression coefficients (β), standard errors (SE), t values, and associated *p*-values. Statistical significance was defined as *p* < 0.05 (*), *p <* 0.001 (***) and *p <* 0.0001 (****).

**Table 6 ijms-26-07929-t006:** Multiple linear regression model examining *APOE–DAOA* interaction effects on AoO.

	Value	Standard Error	T Value	*p*-Value
(Intercept)	40.76	0.68144	59.814	2 × 10^−16^ ****
*DAOA A/A*	3.004	1.0711	2.805	0.00631 **
*DAOA G/A*	0.01273	0.996	0.013	0.98984
*APOEε2+*	−0.26	1.54894	−0.168	0.86712
*APOEε4+*	6.24	1.83484	3.401	0.00105 **
*APOEε2+:G/A*	−2.5127	2.2	−1.14	0.25786
*APOEε4+:G/A*	−2.5127	2.6	−0.964	0.33804
*APOEε2+:A/A*	NA	NA	NA	NA
*APOEε4+:A/A*	−4.25471	2.6366	−1.614	0.11053
Residual standard error: 3.407				
R-square adjusted: 0.2469				
F statistic: 5.074 with 2 and 80 grades of freedom				
*p*-value of F statistic: 8.452 × 10^−5^ ****				

The model included main effects for *DAOA* genotypes (A/A and G/A), *APOE* alleles (*ε2* and *ε4*), and their respective interaction terms. The *DAOA* G/G genotype and *APOEε3/ε3* served as reference categories. Significant associations were identified for the *DAOA* A/A genotype (β = 3.00, *p* = 0.0063) and the presence of *APOEε4* (β = 6.24, *p* = 0.00105), both of which were associated with a later AoO. Interaction terms did not reach statistical significance, indicating no detectable synergistic effects between *DAOA* and *APOE* in this model. Notably, the interaction between *APOEε2* and *DAOA* A/A could not be estimated due to a lack of observations in that subgroup. Statistical significance was defined as *p <* 0.01 (**) and *p <* 0.0001 (****).

## Data Availability

The data supporting the findings of this study can be obtained from the corresponding author upon reasonable request. Access to the data is restricted due to privacy and ethical considerations.

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
