# Peer review of "DAOA and APOEε4 as Modifiers of Age of Onset in Autosomal-Dominant Early-Onset Alzheimer’s Disease Caused by the PSEN1 A431E Variant"

_ijms, 2025, doi:10.3390/ijms26167929_

Round 1

Reviewer 1 Report

Comments and Suggestions for Authors

This study investigates how APOE and DAOA rs2391191 genotypes influence the age of onset (AoO) and first clinical manifestation (FCM) in 88 individuals carrying the PSEN1 A431E mutation, which causes autosomal dominant early-onset Alzheimer’s disease (AD-EOAD). The authors report that both the APOEε4 and DAOA A/A genotypes are associated with a delayed AoO, but not with differences in FCM. The study focuses on a genetically homogeneous cohort, likely due to a founder effect in Jalisco, Mexico, potentially enhancing power to detect genotype-phenotype associations by reducing background genetic variability.

By identifying potential genetic modifiers of AoO, the study holds relevance for risk stratification, genetic counseling, and clinical trial design in AD-EOAD from THIS SPECIFIC AREA.

However, there are several obvious limitations:

  1. While this represents one of the largest known A431E cohorts (n = 88), the study remains underpowered to detect subtle genotype effects, gene-gene interactions, or perform sex-stratified analyses. For instance, there is only one APOEε4/ε4 carrier, limiting any conclusions about dose effects.

  1. The findings are based on a Mexican founder population, which may limit generalizability to other ethnic backgrounds or PSEN1 variants.

  1. "APOEε4 as protective" may be misleading, especially without accounting for known context-dependent effects of APOE alleles. A more nuanced interpretation is warranted.

  1. Although two genetic loci were analyzed, the study did not assess potential interactions between APOE and DAOA, nor did it consider environmental or lifestyle factors that could modify AoO, as this finding contrasts with the existing literature.

  1. There are factual inaccuracies in the background—for example, the statement: “Worldwide, approximately 50 million people have dementia, and this number is projected to increase to 13.8 million by 2060” appears incorrect. The projected global burden is generally expected to exceed 130 million by 2050.

  1. Minor language and grammatical issues were noted (e.g., Line 64: “refer to as” should be corrected for clarity).
Comments on the Quality of English Language

Minor language and grammatical issues were noted.

Author Response

We sincerely thank the Editor for the time and effort dedicated to the review of our manuscript. We are grateful for the constructive feedback, which has helped us improve the clarity and rigor of our work. Below, we provide detailed responses to each of the reviewers’ comments and describe the corresponding changes made to the manuscript.

  1. While this represents one of the largest known A431E cohorts (n = 88), the study remains underpowered to detect subtle genotype effects, gene-gene interactions, or perform sex-stratified analyses. For instance, there is only one APOEε4/ε4 carrier, limiting any conclusions about dose effects.

As suggested, we have added a statement to the “Limitations and Future Directions” section (lines 470–479) acknowledging that, despite representing one of the largest known autosomal dominant early onset Alzheimer’s disease cohorts, the sample size remains limited for conclusions about gene dosage.

Regarding sex-specific analyses, we conducted a Mann-Whitney U test, as shown in Tables 1; however, the results did not reach statistical significance. Additionally, we incorporated Table 6 and Figure 3, which specifically address potential gene-gene interactions between PSEN1 and APOE, providing further insight into their effects.

We also agree that the study may be underpowered to detect subtle genetic effects. However, the rarity of PSEN1 variants limit the feasibility of larger samples. We hope they will contribute to guiding future meta-analyses or collaborative efforts involving multiple cohorts.

  1. The findings are based on a Mexican founder population, which may limit generalizability to other ethnic backgrounds or PSEN1 variants.

This point has been explicitly addressed in section 5.1 Limitations and Future Directions, where we discuss how the population's genetic background may restrict the applicability of our results to other ethnic groups or PSEN1 variants (lines 470–479).

  1. "APOEε4 as protective" may be misleading, especially without accounting for known context-dependent effects of APOE alleles. A more nuanced interpretation is warranted

We have removed all references to APOEε4 as a “protective” factor and instead refer to it solely as a potential modifier of age at symptom onset. However, the discussion included in lines 327–340 regarding possible compensatory mechanisms associated with APOEε4 in the context of autosomal dominant Alzheimer’s disease has been retained, as these hypotheses are supported by previously published evidence and are consistent with findings in other cohorts.

  1. Although two genetic loci were analyzed, the study did not assess potential interactions between APOE and DAOA, nor did it consider environmental or lifestyle factors that could modify AoO, as this finding contrasts with the existing literature.

Table 6 and Figure 3 have been added to explore potential interactions between APOE and DAOA, which are now discussed in the manuscript. Regarding environmental and lifestyle factors, we acknowledge their potential relevance in modifying age of onset; however, it was not feasible to include such variables in this study, as most of the patients presented with severe cognitive impairment at the time of clinical evaluation, limiting the reliability of retrospective data collection.

  1. There are factual inaccuracies in the background—for example, the statement: “Worldwide, approximately 50 million people have dementia, and this number is projected to increase to 13.8 million by 2060” appears incorrect. The projected global burden is generally expected to exceed 130 million by 2050.

Upon review, we realized that the original statement mistakenly referenced projections specific to the United States rather than global estimates. The figure of 13.8 million corresponds to the projected number of people with dementia in the U.S. by 2060. This has now been corrected in the manuscript to accurately reflect the global burden, which is expected to be 135 million by 2050. We appreciate the reviewer’s careful attention to this detail.

  1. Minor language and grammatical issues were noted (e.g., Line 64: “refer to as” should be corrected for clarity).

We have revised several paragraphs throughout the manuscript where we identified opportunities to improve clarity and precision (e.g.: lines 44-48, 52-66, 132-140, 282-295) This includes the sentence on line 64, which has been rephrased for better readability. We believe these edits enhance the overall coherence and accessibility of the manuscript.

Reviewer 2 Report

Comments and Suggestions for Authors

General Comments

This manuscript investigates the potential modifying effects of APOE and DAOA genotypes on the age of onset (AoO) and first clinical manifestation (FCM) in individuals with early-onset Alzheimer’s disease (EOAD) carrying the PSEN1 A431E pathogenic variant. The study presents new insights from a unique Mexican founder population and contributes valuable data on genetic modifiers in autosomal dominant EOAD (AD-EOAD), a field in which empirical evidence is still limited.

Major Points

1- Can the authors provide a more comprehensive discussion of how the observed delayed AoO in APOEε4 carriers aligns, or contrasts, with established findings across different AD-EOAD cohorts? Could population-specific factors (genetic background, environmental exposure…) confound or mediate this effect? How do the authors address the possibility of selection or survival bias?

2- Have the authors considered including relevant covariates such as sex, education, or comorbidities in the regression models for AoO? Could these variables confound the association between genotype and AoO? Given that linear regression assumes independence and homoscedasticity, have model assumptions been tested?

3- How precisely was the age of onset defined in cases where symptoms may have been subtle or retrospectively reported? Were structured clinical instruments or caregiver diaries used to improve accuracy? Could recall bias affect the temporal resolution of AoO?

4- Have the authors explored possible interaction effects (epistasis) between APOE and DAOA genotypes in modifying AoO? Is there statistical power to detect such interactions, or could these factors act synergistically in certain genotype combinations?

5- While the DAOA findings are biologically plausible, have the authors sought external validation in other EOAD cohorts or functional data (D-serine levels, imaging biomarkers…) that could support the mechanistic claims?

6- Could the authors consider incorporating or at least discussing additional clinical endpoints (rate of progression, MMSE decline, behavioral symptoms)? Do these modifiers influence disease severity or trajectory beyond AoO?

7- The sample size is moderate (n=88). Have the authors conducted post-hoc power analyses to determine the robustness of their genotype-AoO associations? Are the subgroup comparisons (ε4/ε4…) underpowered?

Minor Points

  1. In some places, the manuscript refers to "APOEε2+" and "APOEε4+.” Could the authors clarify whether these include both heterozygous and homozygous individuals? Consistent nomenclature would improve clarity.
  2. The manuscript occasionally refers to “trends” or “suggestions” when p-values are non-significant (APOEε2 protective effect). Could the authors temper such interpretations and state more clearly when findings do not reach statistical significance?
  3. Although ethics approval is mentioned, could the authors clarify how data from deceased individuals or historical family data were handled, particularly for retrospective clinical reconstructions?

This manuscript offers a valuable contribution to the understanding of genetic modifiers in AD-EOAD, particularly in a rare founder population. However, the authors should address several interpretative and methodological limitations before publication. Key improvements include a more nuanced interpretation of APOEε4 effects, consideration of covariates in modeling, and a critical discussion of generalizability and potential confounding.

Author Response

We would like to thank the Reviewer for their thoughtful and constructive comments. Their insights have been extremely helpful in improving the clarity, interpretation, and overall quality of our manuscript. Below, we address each point in detail and outline the corresponding revisions made.

  1. Can the authors provide a more comprehensive discussion of how the observed delayed AoO in APOEε4 carriers aligns, or contrasts, with established findings across different AD-EOAD cohorts? Could population-specific factors (genetic background, environmental exposure…) confound or mediate this effect? How do the authors address the possibility of selection or survival bias?

We have expanded the discussion of how the delayed age of onset observed in APOEε4 carriers aligns or contrasts with previous findings across different AD-EOAD cohorts. We have incorporated this discussion into the revised Discussion section (lines 291–315). We also consider the potential role of population-specific factors, such as genetic background or environmental exposures. However, the assessment of environmental factors in our cohort is limited, as most patients were already in advanced stages of the disease at the time of evaluation, making it difficult to reliably reconstruct exposure histories retrospectively.

Regarding the possibility of selection or survival bias, we believe this is unlikely to have significantly influenced our results. PSEN1 A431E carriers typically remain symptomatic for years—often up to seven—before death, and the earliest reported age at symptom onset in our cohort was 34. It is therefore improbable that individuals with earlier onset were systematically excluded due to premature mortality.

  1. Have the authors considered including relevant covariates such as sex, education, or comorbidities in the regression models for AoO? Could these variables confound the association between genotype and AoO? Given that linear regression assumes independence and homoscedasticity, have model assumptions been tested?

After careful consideration, we decided not to include these variables in our primary regression analysis for the following reasons:

First, our main objective was to assess the direct association between APOE and DAOA genotype and age of onset within a genetically homogeneous cohort of PSEN1 A431E variant carriers. Given the autosomal dominant nature of the mutation, the variability in AoO is presumed to be primarily driven by genetic modifiers such as APOE, rather than by traditional sporadic AD risk factors.

Second, although sex, education, and comorbidities may influence clinical presentation in sporadic AD, their role in the context of monogenic, early-onset familial AD is less clear and likely to be marginal. Including them as covariates could introduce unnecessary noise into the model, especially considering our limited sample size.

Finally, preliminary subgroup analyses (e.g., Mann-Whitney U test for sex differences; see Table 1) did not reveal significant differences with AoO. As such, we believe that including these variables as covariates would not meaningfully improve the model and might reduce statistical power due to overfitting.

We tested the key assumptions of the linear regression model. The Shapiro-Wilk test indicated that the residuals follow an approximately normal distribution (W = 0.973, p = 0.064), and the Breusch-Pagan test showed no evidence of heteroscedasticity (BP = 3.63, df = 4, p = 0.459), supporting the assumption of homoscedasticity. These results suggest that the use of a linear regression model is appropriate for our data.

3- How precisely was the age of onset defined in cases where symptoms may have been subtle or retrospectively reported? Were structured clinical instruments or caregiver diaries used to improve accuracy? Could recall bias affect the temporal resolution of AoO?

Age of onset was defined as the age at which the first cognitive or functional symptoms were clearly reported by the patient or caregiver and subsequently documented in the clinical record. In cases where symptoms were retrospectively reported, detailed clinical interviews with caregivers and neurologists were used to estimate onset, often triangulated across multiple visits and corroborated by functional scales or caregiver reports. Although structured instruments or diaries were not consistently used across all participants, efforts were made to standardize this assessment as much as possible.

  1. Have the authors explored possible interaction effects (epistasis) between APOE and DAOA genotypes in modifying AoO? Is there statistical power to detect such interactions, or could these factors act synergistically in certain genotype combinations?

We conducted an exploratory analysis to examine the potential interaction between APOE and DAOA genotypes in relation to age of onset. The results of this analysis are now presented in Table 6 of the revised manuscript.

  1. While the DAOA findings are biologically plausible, have the authors sought external validation in other EOAD cohorts or functional data (D-serine levels, imaging biomarkers…) that could support the mechanistic claims?

Notably, Vélez et al. (DOI: 10.1155/2016/9760314) reported a similar association between DAOA and age at onset in carriers of the PSEN1 E280A variant (a larger AD-EOAD cohort in Colombia), suggesting that DAOA may act as a modifier across different AD-EOAD populations. Based on these converging findings, we hypothesize that the biological mechanism may involve the modulation of glutamatergic signaling and D-serine metabolism by DAOA, consistent with its proposed role in NMDA receptor regulation.

While we did not have access to functional data (e.g., D-serine levels, neuroimaging biomarkers) in our cohort, we acknowledge this limitation and have explicitly stated in the revised Limitations section that functional studies are needed to confirm the mechanistic implications of our results and to validate DAOA as a potential genetic modifier in AD

6- Could the authors consider incorporating or at least discussing additional clinical endpoints (rate of progression, MMSE decline, behavioral symptoms)? Do these modifiers influence disease severity or trajectory beyond AoO?

We agree with the reviewer that assessing additional clinical endpoints such as rate of progression, MMSE decline, or behavioral symptoms would provide valuable insights into whether genetic modifiers like APOE and DAOA influence disease severity or trajectory beyond age of onset. However, in our cohort, this type of analysis was not feasible. In most cases, patients presented to the clinic at advanced stages of the disease, often with significant cognitive and functional impairment already established. As a result, baseline cognitive scores and longitudinal follow-up data were either unavailable or not suitable for reliable modeling of disease progression.

We have now noted this limitation in the Limitations section and emphasized the importance of future studies with prospective follow-up and earlier-stage recruitment to evaluate the potential impact of these genetic variants on disease severity and progression.

  1. The sample size is moderate (n=88). Have the authors conducted post-hoc power analyses to determine the robustness of their genotype-AoO associations? Are the subgroup comparisons (ε4/ε4…) underpowered?

We acknowledge the reviewer’s concern regarding the sample size and its potential impact on the robustness of genotype–AoO associations, particularly for rare APOE genotypes such as ε4/ε4. To address this, we have now clarified in the revised manuscript (line 174-181) that APOE genotypes were grouped into broader subgroups: APOEε2+, APOEε3+, and APOEε4+, in line with previous literature, to increase statistical power while maintaining biological relevance.

Furthermore, we performed a post-hoc power analysis for the multiple linear regression model used to assess the association between APOE and DAOA genotypes and AoO. The analysis yielded a statistical power of 0.9458, indicating sufficient power to detect moderate effect sizes within the constraints of our sample. This analysis supports the validity of our findings, while we continue to interpret subgroup comparisons with appropriate caution due to limited representation of rare genotypes.

Minor Points

  1. In some places, the manuscript refers to "APOEε2+" and "APOEε4+.” Could the authors clarify whether these include both heterozygous and homozygous individuals? Consistent nomenclature would improve clarity.

To address this, we have now clarified in the revised manuscript (line 174-181) that APOE genotypes were grouped into broader subgroups: APOEε2+, APOEε3+, and APOEε4+, in line with previous literature, to increase statistical power while maintaining biological relevance.

  1. The manuscript occasionally refers to “trends” or “suggestions” when p-values are non-significant (APOEε2 protective effect). Could the authors temper such interpretations and state more clearly when findings do not reach statistical significance?

The protective effect mentioned in the manuscript refers to the APOEε2 allele in the sporadic form of Alzheimer’s disease, for which a protective role has been widely documented in the literature (e.g., lines 88–89 and 454). Nevertheless, we have revised the manuscript to ensure that our interpretation does not overstate the findings. This adjustment aims to temper the interpretation and maintain scientific rigor

  1. Although ethics approval is mentioned, could the authors clarify how data from deceased individuals or historical family data were handled, particularly for retrospective clinical reconstructions?

We confirm that no post-mortem samples were collected for this study. All biological samples and clinical data were obtained while participants were alive, and informed consent was provided either by the individuals themselves or, when applicable, by their legal caregivers. The consent form explicitly stated that the collected samples and associated data could be used for the present study as well as for future research purposes. For retrospective clinical reconstructions, we relied exclusively on data obtained during the participants’ lifetime, ensuring full compliance with ethical standards. The study protocol was reviewed and approved by the institutional ethics committee (Approval ID: R-2022-1305-100).

Round 2

Reviewer 1 Report

Comments and Suggestions for Authors

The authors have made efforts to improve this manuscript based on the reviewers’ comments. However, the statement that “ε3 was the most prevalent (82.02%), followed by ε2 (10.11%) and ε4 (7.87%)” appears to be inaccurate upon recalculation of the percentages.

Author Response

We appreciate the reviewer’s careful attention to detail and for identifying the issue regarding reported percentages. The statement has been corrected accordingly in the revised manuscript, and we have performed an additional verification to ensure that similar errors are not present in other sections of the text.

Reviewer 2 Report

Comments and Suggestions for Authors

The authors have successfully incorporated the suggested revisions and clarified the outstanding issues. I recommend the manuscript for publication.

Author Response

We thank the reviewer for the positive feedback and recommendation for publication. To further improve the introduction, we have added lines 79–86, which provide additional context on the APOE alleles.